# Electromechanical coupling mechanism for activation and inactivation of an HCN channel

Gucan Dai [1✉], Teresa K. Aman[1], Frank DiMaio[2] & William N. Zagotta [1✉]

Pacemaker hyperpolarization-activated cyclic nucleotide-gated (HCN) ion channels exhibit a reversed voltage-dependent gating, activating by membrane hyperpolarization instead of depolarization. Sea urchin HCN (spHCN) channels also undergo inactivation with hyperpolarization which occurs only in the absence of cyclic nucleotide. Here we applied transition metal ion FRET, patch-clamp fluorometry and Rosetta modeling to measure differences in the structural rearrangements between activation and inactivation of spHCN channels. We found that removing cAMP produced a largely rigid-body rotation of the C-linker relative to the transmembrane domain, bringing the A' helix of the C-linker in close proximity to the voltage-sensing S4 helix. In addition, rotation of the C-linker was elicited by hyperpolarization in the absence but not the presence of cAMP. These results suggest that — in contrast to electromechanical coupling for channel activation — the A' helix serves to couple the S4-helix movement for channel inactivation, which is likely a conserved mechanism for CNBD-family channels.

[1] Department of Physiology and Biophysics, University of Washington, Seattle, WA, USA. [2] Department of Biochemistry, University of Washington, Seattle, WA, USA. ✉email: daig@uw.edu; zagotta@uw.edu

Hyperpolarization-activated cyclic nucleotide-gated (HCN) ion channels are opened by membrane hyperpolarization, a phenomenon important for pacemaking in the heart as well as for spontaneous neuronal firing, oscillatory activity, and synaptic transmission in the brain[1–3]. This type of electro-mechanical coupling is reversed compared to most other voltage-gated ion channels (VGICs), which are activated by membrane depolarization[4,5]. Paradoxically, HCN channels share a common architecture with other VGICs—particularly cyclic nucleotide-gated (CNG) and KCNH channels—with four similar or identical subunits around a centrally-located pore. Each subunit consists of a voltage-sensing domain (VSD) with four (S1–S4) transmembrane helices, a pore domain (PD) with S5–S6 transmembrane helices, an amino-terminal HCN domain (HCND), a carboxy-terminal C-linker domain, and a carboxy-terminal cyclic nucleotide-binding domain (CNBD) (Fig. 1a)[3,6,7]. Distinct from the "domain-swapped" VSD-PD architecture of canonical VGICs[8], however, the VSD and PD of CNBD-family channels, including HCN, CNG, and KCNH channels, are not domain swapped (Fig. 1b)[9–11]. The VSD of "non-domain-swapped" channels is adjacent to the PD of the same subunit so that the S4 helix is closely packed to the S5 helix of the same subunit but also in close proximity to the S6 helix and the C-linker region of the neighboring subunit (Fig. 1a, b). Interestingly, CNBD family channels, like HCN and KCNH channels, the VSD and C-linker are in a VSD/C-linker domain-swapped configuration (Fig. 1b). Despite these structural similarities between HCN channels and KCNH channels (e.g., EAG1 and hERG1) (Supplementary Fig. 1a), HCN channels are activated by membrane hyperpolarization whereas KCNH channels are activated by depolarization (Supplementary Fig. 1b). Sea urchin sperm HCN (spHCN) channels also undergo a voltage-dependent inactivation process with hyperpolarization, making the channels opened by both hyperpolarization (activation) and depolarization (recovery from inactivation)[12–14].

The detailed movement of the S4 helix of spHCN has been characterized using transition metal ion Förster resonance energy transfer (tmFRET) combined with Rosetta modeling[15]. Upon hyperpolarization, the S4 helix exhibits an ~10 Å downward translation and a tilting/bending motion in the C-terminal portion of the helix[15]. Two voltage-sensing arginines in the S4 move from above the hydrophobic constriction site (HCS), a narrow region in the center of the VSD characterized by a conserved hydrophobic residue (phenylalanine in spHCN)[5], to below the HCS in response to −100 mV hyperpolarization. This movement likely underlies the voltage-dependence of the hyperpolarization-dependent activation[15]. This rearrangement is consistent with a recent cryo-electron microscopy (cryo-EM) structure of HCN1 locked in the activated position by a metal bridge[16], as well as molecular dynamics simulations[17].

Electromechanical coupling is the process by which changes in membrane voltage produce a rearrangement of the VSD that is coupled to a rearrangement of the PD. The polarity of voltage-dependent gating is likely determined by the direction of the coupling between the VSD and the PD. The rearrangement of the VSD of HCN channels in response to voltage is generally similar to that in other depolarization-activated channels, with the positively-charged S4 helix moving downward upon hyperpolarization (upward upon depolarization)[15–20]. Interestingly, minimal mutations in the intracellular sides of the S4 and S5 helices could reverse the gating polarity of HCN channels, making them depolarization activated[14,21,22]. Some mutations produce channels that are both hyperpolarization activated and depolarization activated[14,22], suggesting that the coupling pathways for hyperpolarization and depolarization activation are distinct.

The mechanism of electromechanical coupling is thought to differ between the domain-swapped VGICs and the non-domain-swapped CNBD channels. For these domain-swapped channels, the coupling is thought to occur primarily through the covalent linkage between the S4 helix of the VSD and the S5 helix of the PD, the S4–S5 linker[5,20,23]. This S4–S5 linker-dependent pathway is considered the canonical electromechanical coupling for VGICs. However, a covalent linkage between the S4 and S5 segment is not required for the voltage-dependent gating of the non-domain-swapped channels EAG1, hERG1, and spHCN[14,24,25]. For these non-domain-swapped channels, the electromechanical coupling has been proposed to involve either a noncovalent interaction between the S4 and S5 helices of the same subunit or via the C-linker in the adjacent subunit[22,26].

Cyclic nucleotide monophosphates (cNMPs) directly bind and potentiate the activity of HCN channels, an important pathway for regulating HCN channels during cell signaling[4,27]. For most HCN channels, cyclic adenosine monophosphate (cAMP) is a full agonist for the CNBD and promotes channel opening[28], causing a depolarizing shift in the voltage dependence of activation. In spHCN channels, however, cAMP also eliminates the voltage-dependent inactivation that occurs in the absence of cyclic nucleotide. Binding of cAMP to the CNBD induces a conformational change that is relayed to the transmembrane regions of HCN channels via the C-linker[3,29,30]. The C-linker region is comprised of six helices: A'–F' helices, with the A' helix immediately following the S6 transmembrane helix and running nearly parallel to the membrane[6,27]. The A' and B' helices form a helix-turn-helix motif that interacts with the C' and D' helices of the neighboring subunit. This intersubunit "elbow-on-the-shoulder" interaction forms a gating ring immediately below the transmembrane region and is involved in the cyclic nucleotide-dependent regulation of HCN and CNG channels[29].

Here we used tmFRET to determine the conformational changes of spHCN channels in response to membrane hyperpolarization and cyclic-nucleotide binding. Distances measured with tmFRET under different conditions—in the presence and absence of cAMP, and at different voltages (0 or −100 mV)—were used as constraints for Rosetta-based structural modeling. The changes in distance could be accounted for by a largely rigid-body rotation of the C-linker relative to the transmembrane domain, putting the A' helix in close proximity to the S4 helix in the absence of cyclic nucleotide. We propose a mechanism where a rearrangement of the A' helix in the C-linker could help to translate the downward movement of the S4 into closure of the channel pore (inactivation). In the presence of cAMP, the A' helix moves further away from the S4 helix, permitting the hyperpolarization-dependent activation mediated by the S5 helix.

## Results

**Inactivation of spHCN channels in the absence of cyclic nucleotide.** Cyclic nucleotides are required to maintain the hyperpolarization-dependent opening of spHCN channels[12]. In the presence of 1 mM cAMP, spHCN channels were activated by hyperpolarizing voltage pulses, similar to mammalian HCN channels (Fig. 1c). However, in the absence of cyclic nucleotide, the spHCN channels rapidly inactivated and exhibited much smaller peak currents compared to those in the presence of cAMP[13] (Fig. 1c). Previously it has been shown that this inactivation process is primarily from closed states and appears to be voltage dependent, occurring more rapidly and completely at hyperpolarized voltages[13,14,31]. In addition, removing the carboxy-terminal region of spHCN channels, including the CNBD and C-linker, eliminates this inactivation[32,33]. The findings that inactivation is voltage dependent and eliminated by

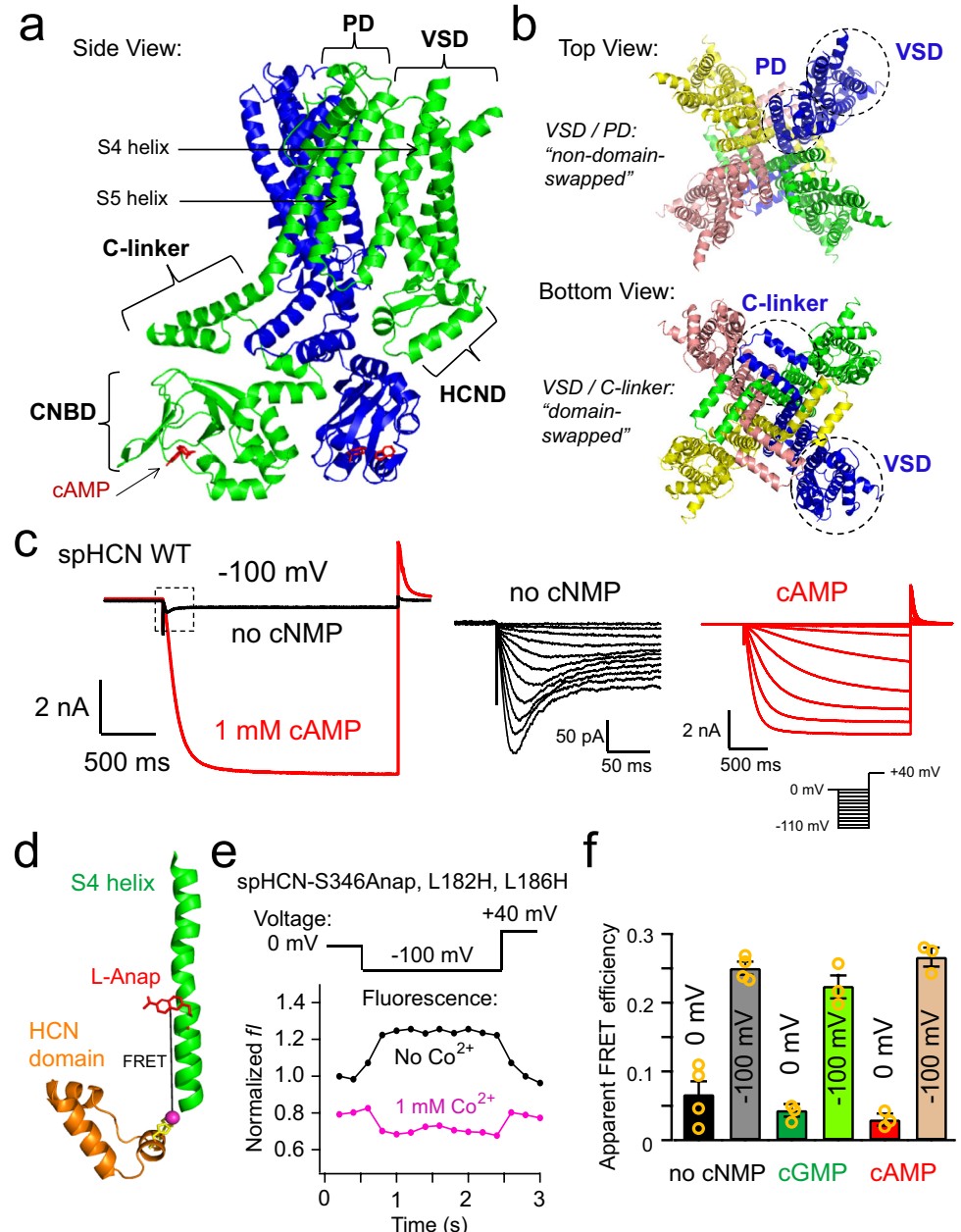

**Fig. 1 Structural features and properties of spHCN channels in the presence and absence of cyclic nucleotide. a** Side view of a homology structure of spHCN channels based on the cryo-EM structure of human HCN1 (PDB 5U6P), highlighting the proximity of the S4 and S5 transmembrane helices relative to the C-linker domain of the adjacent subunit. Only two adjacent subunits are shown. **b** Top and bottom views of the same homology model of spHCN showing the arrangement of its quaternary structure. **c** Left: spHCN channel currents elicited by a hyperpolarizing voltage pulse from 0 to −100 mV, in the presence (red) or in the absence (black) of 1 mM cAMP. Middle: Channel inactivation without cyclic nucleotide, elicited by a series of hyperpolarizing voltages from 0 to −120 mV (with 10 mV steps), happened during the time frame indicated by the dashed box on the left panel. Note the blown up current and time scale. Right: Channel currents elicited by a series of hyperpolarizing voltages from 0 to −120 mV (with 10 mV steps), in the presence of 1 mM cAMP. **d** Left, cartoon showing the FRET between S346Anap in the S4 helix and $Co^{2+}$ chelated by a di-histidine L182H and L186H in the HCN domain. **e** Fluorescence measurements for spHCN-S346Anap, L182H, and L186H channels with a −100 mV voltage pulse, before and after applying 1 mM $Co^{2+}$. **f** Summary of the FRET efficiencies measured for spHCN-S346Anap, L182H, and L186H channels at 0 and −100 mV in the absence of cyclic nucleotide monophosphate (no cNMP; $n = 4$ independent patches), and the presence of cGMP ($n=3$ independent patches) or cAMP ($n=3$ independent patches). Data shown are mean ± s.e.m. Source data are provided in the Source Data file.

cAMP or carboxy-terminal deletion suggest that the spHCN inactivation is tightly coupled to voltage-sensor movement and involves the carboxy-terminal region, particularly the C-linker.

**Voltage-sensor movement in the absence of cyclic nucleotide.** To understand the structural mechanism of the inactivation of

spHCN channels, we asked whether the S4 helix in the VSD still moved with hyperpolarization in the absence of cyclic nucleotide when the channels are largely inactivated. As we previously showed[15], the fluorescent noncanonical amino acid L-Anap, with its environmentally-sensitive fluorescence emission, allows us to observe the movement of the S4 by measuring the change in

environment of S4 residues upon hyperpolarization. We introduced L-Anap at Ser[346], in the middle of the S4 helix of a spHCN-YFP fusion construct, using the *amber* stop-codon (TAG) suppression strategy in *Xenopus* oocytes[15] (Supplementary Fig. 2a). Patch-clamp fluorometry (PCF) was used to simultaneously measure the fluorescence and ionic current from giant inside-out patches from oocytes, while controlling membrane voltage and rapidly applying intracellular ligands (e.g., cAMP and transition metals)[34,35]. Specific L-Anap incorporation and full-length channel expression were confirmed by the correlation of the magnitude of Anap fluorescence with both the YFP fluorescence and spHCN ionic currents (see[15]).

As we reported previously, there was a substantial increase in the Anap fluorescence in spHCN-S346Anap during hyperpolarizing voltage pulses of −100 mV in the presence of a saturating concentration (1 mM) of the full agonist cAMP ($\Delta F/F = 61.2 \pm 3.3\%$) (Supplementary Fig. 2b)[15]. The partial agonist cyclic guanosine monophosphate (cGMP) produced much smaller ionic current, and the absence of cyclic nucleotide generated negligible current, with steps to −100 mV; however, the Anap fluorescence still elevated considerably ($\Delta F/F = 46.3 \pm 2.2\%$ in cGMP and $\Delta F/F = 26.0 \pm 3.3\%$ in the absence of cyclic nucleotide) (Supplementary Fig. 2b). These results suggest that the S4 helix moved with hyperpolarizing voltage regardless of whether the channel is inactivated (in the absence of cyclic nucleotide) or activated (in the presence of cAMP). Nevertheless, use of the environmental sensitivity of L-Anap provided limited structural information about the size of the S4 movement.

To measure the voltage-sensor movement more quantitatively, we used tmFRET[36,37]. tmFRET measures the distance between a donor fluorophore and an acceptor non-fluorescent transition metal ion, such as $Ni^{2+}$, $Co^{2+}$, and $Cu^{2+}$, bound to minimal transition metal ion binding sites introduced into the protein. Transition metals such as $Ni^{2+}$, $Co^{2+}$, and $Cu^{2+}$ have absorption spectra that overlap with the emission spectrum of L-Anap and hence can serve as nonfluorescent FRET acceptors that quench the donor's fluorescence in a highly distance-dependent manner. Because the absorption of most transition metals is low, with multiple transition dipoles, tmFRET can measure short distances (10–25 Å) with little or no orientation dependence. L-Anap and metal bound to an introduced di-histidine motif are closely associated with the protein backbone and well suited as a tmFRET pair for measuring the backbone distances and changes in distance associated with protein conformational changes.

To quantify the downward movement of the S4 helix, we measured tmFRET between S346Anap in the S4 and $Co^{2+}$ bound to a di-histidine motif (L182H–L186H) introduced into an α helix of the amino-terminal HCND directly below the S4, as previously described (Fig. 1d)[15]. Upon application of 1 mM $Co^{2+}$, there was substantial quenching of Anap fluorescence indicative of FRET between S346Anap and $Co^{2+}$ bound to the di-histidine site in the HCND. In the presence of 1 mM $Co^{2+}$, the Anap fluorescence was decreased by the −100 mV hyperpolarization, instead of increased in the absence of $Co^{2+}$, indicating that the quenching (and therefore FRET efficiency) was greater at −100 mV than at 0 mV (Fig. 1e). We quantified the apparent tmFRET efficiency at each voltage by calculating the fractional decrease in Anap fluorescence produced by 1 mM $Co^{2+}$ and correcting for the solution quenching in spHCN-S346Anap channels lacking the di-histidine site[15]. In the absence of cyclic nucleotide, FRET efficiency increased substantially at −100 mV, similar to the increase seen in cAMP and cGMP (Fig. 1f). These results indicate that the Ser[346] position in the S4 helix moved downward–closer to the HCND—with hyperpolarization in the absence of cyclic nucleotide, similar to the movement with cAMP or cGMP. These data suggest that inactivation of spHCN channels in the absence

of cyclic nucleotide does not arise from immobilization or a fundamentally different movement of the S4 helix.

**Rearrangement of the C-linker relative to the S4 helix.** How does the spHCN channel inactivate in the absence of cyclic nucleotide? The C-linker is situated just below the transmembrane domain and directly communicates cAMP binding to the CNBD to the transmembrane domain. Therefore, we studied the movement of the C-linker relative to the transmembrane domain using tmFRET[15,36,38,39]. For these and subsequent experiments in this study, we used a variation of tmFRET called ACCuRET (Anap Cyclen-$Cu^{2+}$ resonance energy transfer)[15,36,38,39]. ACCuRET measures the FRET efficiency between a donor fluorophore (L-Anap) and an acceptor $Cu^{2+}$ chelated by TETAC (1-(2-pyridin-2-yldisulfanyl) ethyl)-1,4,7,10-tetraazacyclododecane)[15,39]. TETAC is a cysteine-reactive compound with a short linker to a cyclen ring that binds transition metal ions with subnanomolar affinity. Single cysteines introduced into the protein react with $Cu^{2+}$-TETAC to introduce a transition metal ion acceptor closely associated with the protein backbone. This configuration of tmFRET allowed us to measure the FRET efficiency as the fractional quenching of Anap fluorescence upon modification by $Cu^{2+}$-TETAC. The distance between Anap and $Cu^{2+}$-TETAC could then be calculated from the FRET efficiency.

ACCuRET was used to measure distances between the transmembrane domain and the C-linker of spHCN channels in the presence and absence of cAMP, at 0 and −100 mV. For the tmFRET donor, L-Anap was incorporated into a cysteine-reduced spHCN-YFP construct at Trp[355] (spHCN-W355Anap), near the intracellular end of the S4 helix, using amber stop-codon suppression[15]. This cysteine-reduced spHCN channel behaved similarly to wild-type channels as previously reported[15]. In addition, we have previously demonstrated efficient and specific L-Anap labeling at the W355 position in spHCN[15]. For the tmFRET acceptors, single cysteines were introduced into the C-linker at two different sites, L481C in the A′ helix (Fig. 2a) and E506C in the B′ helix (Fig. 2b). Distances between the donor and acceptor were predicted to be much shorter for adjacent subunits (22.1 Å for L481 and 18.8 Å for E506) than for the same subunit (41 Å for L481 and 44.8 Å for E506) or between diagonal subunits (28.1 Å for L481 and 35.4 Å for E506) based on the β-carbon distances in the solved hHCN1 structure (PDB code: 5U6P)[9]. Therefore, considering the $R_0$ is around 13 Å for W355Anap (see methods), tmFRET is expected to occur primarily between donors and acceptors in adjacent subunits for these sites (also see Supplementary Table 1 summarizing the distance measurements of final structural models). These channels, with W355Anap and $Cu^{2+}$-TETAC modification of the C-linker sites, exhibited hyperpolarization-dependent activation in the presence of cAMP and inactivation in the absence of cAMP similar to the wild-type channel (Supplementary Fig 3a, c).

As above, PCF was used to simultaneously measure the tmFRET and ionic current while controlling membrane voltage and intracellular ligands (cAMP and $Cu^{2+}$-TETAC)[34,35] (Supplementary Fig. 3). For spHCN-W355Anap, L481C, the Anap fluorescence was significantly quenched after 1-min application of 20 μM $Cu^{2+}$-TETAC to the intracellular side of the patch. The degree of quenching (1 − $F_{Cys}$) was considerably higher than the nonspecific quenching (1 − $F_{no\ Cys}$) seen for spHCN-W355Anap without L481C (Fig. 2c). At 0 mV, the $Cu^{2+}$-TETAC quenching of Anap fluorescence was greater in the absence of cyclic nucleotide than in the presence of cAMP (Fig. 2c and Supplementary Fig. 4a). With voltage steps to −100 mV, Anap fluorescence decreased even in the control condition (in the absence of $Cu^{2+}$-TETAC) due to the environmental change

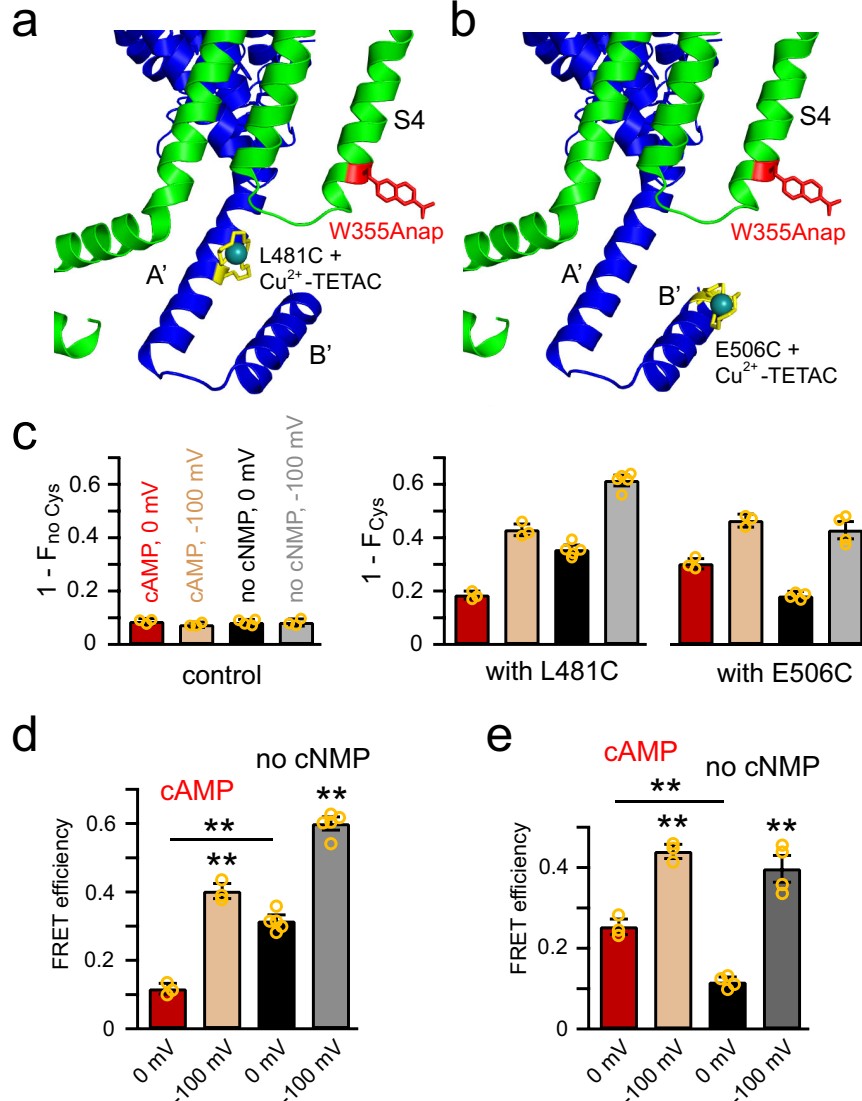

**Fig. 2 tmFRET between the C-terminal end of the S4 helix and A′ or B′ helix of the C-linker. a** Cartoon showing tmFRET between W355Anap in the S4 helix and $Cu^{2+}$-TETAC attached to L481C in the A′ helix. **b** Cartoon showing tmFRET between W355Anap and $Cu^{2+}$-TETAC attached to E506C in the B′ helix. **c** Summary of the fraction of Anap fluorescence quenched by $Cu^{2+}$-TETAC for the spHCN-W355Anap channels in panels **a** and **b**, without (left) and with (middle, right) the introduced cysteines L481C or E506C in four conditions: the presence and absence of cAMP, and at 0 and −100 mV; $n = 3$–5 independent patches. Data shown are mean ± s.e.m.; Source data in the forms of $F_{Cys}$, and $F_{no\ Cys}$ are provided in the Source Data file. **d** FRET efficiencies of spHCN-W355Anap, L481C channels measured in four conditions: the presence and absence of cAMP, and at 0 and −100 mV; $n = 3$ independent patches for cAMP conditions and $n = 5$ independent patches for no cNMP conditions. **p** $< 0.001$ for the statistical comparisons. **e** FRET efficiencies of spHCN-W355Anap, E506C channels; $n = 3$ patches for cAMP and $n = 4$ patches for no cNMP conditions. **p** $< 0.001$ for the statistical comparisons. Data shown are mean ± s.e.m.; source data are provided in the Source Data file.

associated with the movement of W355Anap, as we previously reported[15] (Supplementary Fig. 4a). However, at −100 mV, $Cu^{2+}$-TETAC produced an additional decrease in fluorescence, greater than that at 0 mV, and again greater in the absence of cyclic nucleotide than in the presence of cAMP. FRET efficiencies were calculated from the fractional quenching in Anap fluorescence after application of $Cu^{2+}$-TETAC for each condition (0 and −100 mV, in the presence and absence of cAMP) and corrected for the small $Cu^{2+}$-TETAC-dependent quenching in the control lacking L481C. Since the FRET efficiency was measured independently at 0 and −100 mV, the measurement was not affected by the environmental change in Anap fluorescence. Accordingly, at both 0 and −100 mV, the absence of cyclic nucleotide yielded a higher FRET efficiency than the presence of

cAMP, indicating a shorter distance between $Trp^{355}$ in the S4 and $Leu^{481}$ in the A′ helix in the absence of cyclic nucleotide. At −100 mV, the FRET efficiencies all increased significantly relative to 0 mV, consistent with the downward movement of S4 helix with hyperpolarization (Fig. 2c, d and Supplementary Fig. 4a).

Similar experiments were done on spHCN-W355Anap, E506C with the $Cu^{2+}$-TETAC site relocated to the B′ helix in the C-linker (Fig. 2c, e). As above, the channel without the introduced cysteine (spHCN-W355Anap) was used as the control for correcting for any nonspecific quenching to calculate FRET efficiency. Surprisingly, with the acceptor at the B′ helix site, at both 0 and −100 mV, the FRET efficiency was less in the absence of cyclic nucleotide than in the presence of cAMP, opposite to what we observed at the A′ helix site (Supplementary Fig. 4b). Furthermore, as expected, the FRET

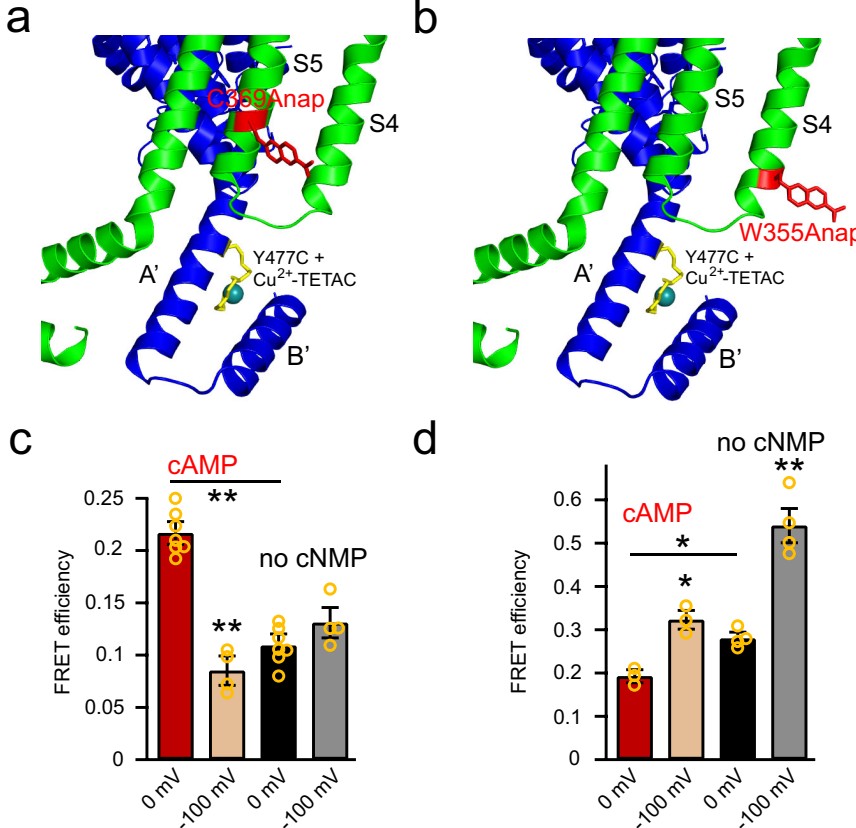

**Fig. 3 tmFRET between the N-terminal end of the S5 helix and A′ helix of the C-linker. a** Cartoon showing tmFRET between C369Anap in the S5 helix and Cu$^{2+}$-TETAC attached to Y477C in the A′ helix. **b** Cartoon showing tmFRET between W355Anap in the S4 helix and Cu$^{2+}$-TETAC attached to Y477C in the A′ helix. **c** FRET efficiencies of spHCN-C369Anap, Y477C channels measured in four conditions: the presence and absence of cAMP, and at 0 and −100 mV; $n = 7$ independent patches for 0 mV conditions and $n = 4$ independent patches for −100 mV conditions. **p < 0.001; $p = 0.1$ between no cNMP, 0 mV and no cNMP, −100 mV conditions. **d** FRET efficiencies of spHCN-W355Anap, Y477C channels measured in four conditions: the presence and absence of cAMP, and at 0 and −100 mV; $n = 3$ independent patches for cAMP and $n = 4$ independent patches for no cNMP. **p < 0.001 and *p < 0.05. Data shown are mean ± s.e.m.; source data are provided in the Source Data file.

efficiency increased with hyperpolarization due to the downward movement of the S4 helix. Together, these results reveal a rearrangement of the C-linker in the absence of cyclic nucleotide, moving the A′ helix closer to the S4 helix and the B′ helix further from the S4 helix. As we discuss below, this rearrangement may permit the inactivation process observed, in the absence of cyclic nucleotide, in spHCN channels.

**Rearrangement of the A′ helix of the C-linker relative to the S5 helix.** Next, we studied the movement of the C-linker relative to the S5 helix by incorporating L-Anap into position Cys$^{369}$ near the intracellular end of the S5 helix. The spHCN-C369Anap channel expressed well, produced a considerable amount of current in the presence of cAMP (Supplementary Fig. 5a), and exhibited a strong correlation between the fluorescence intensities of L-Anap emission and the ionic current and emission of the C-terminal YFP (Supplementary Fig. 5b). In contrast to channels with L-Anap incorporated into the S4 helix, the spHCN-C369Anap channels did not exhibit a noticeable environmental change of Anap fluorescence between 0 and −100 mV (Supplementary Fig. 5c).

We then measured tmFRET between Anap on the S5 helix (C369Anap) and Cu$^{2+}$-TETAC on a site in the A′ helix (Y477C) (Fig. 3a) and compared it to the tmFRET between Anap on the S4 helix (W355Anap) and Cu$^{2+}$-TETAC on the same A′ helix site (Y477C) (Fig. 3b). The spHCN-C369Anap, Y477C channels show

similar cAMP modulation, inactivation in the absence of cyclic nucleotide, and voltage dependence of activation, before and after applying Cu$^{2+}$-TETAC, to the wild-type channels (Supplementary Fig. 6). Note that in these experiments, Y477C was used on the A′ helix instead of L481C (one helix turn away) because Y477C is closer to the Cys$^{369}$ position of the S5 helix. Indeed, for spHCN-W355Anap, Y477C channels, the results for tmFRET were quite similar to those from spHCN-W355Anap, L481C channels (Fig. 3d).

For these two tmFRET pairs with donor sites on the S5 and S4 helices, cAMP produced opposite effects at 0 mV. When L-Anap was located in the S5 helix, the FRET efficiency was lower in the absence of cyclic nucleotide than in the presence of cAMP (Fig. 3c). Conversely, when L-Anap was in the S4 helix, the FRET efficiency was higher in the absence of cyclic nucleotide than in the presence of cAMP (Fig. 3d). These results suggest that the absence of cyclic nucleotide produced a rearrangement of the A′ helix at 0 mV, bringing the A′ helix closer to the S4 helix and further away from the S5 helix.

In the presence of cAMP, applying the −100 mV hyperpolarizing pulse produced a strikingly lower FRET efficiency for spHCN-C369Anap, Y477C (Cu$^{2+}$-TETAC) channels (Fig. 3c). This increase in distance likely results from a movement of S5 helix coupled to the movement of the S4 at −100 mV. Such a movement of the S5 has recently been demonstrated in a cryo-EM structure of the down state of HCN1[16] and can quantitatively account for our observed decrease in FRET efficiency (see

modeling below). Interestingly, in the absence of cyclic nucleo-tide, −100 mV pulses failed to produce this decrease in FRET (Fig. 3c); instead, increased FRET efficiency slightly. This difference in voltage-dependence of the S5-A′ helix distance in the presence and absence of cAMP is likely related to the inactivation of spHCN in the absence of cAMP. It could result from either a lack of S5 helix movement in the inactivated state, or a difference in the movement of the A′ helix in the inactivated state, or both (see modeling below).

**Rosetta models for the cAMP- and hyperpolarization-dependent gating.** To perform a more comprehensive structural analysis of our FRET data in various states, we used our FRET efficiencies to determine the distances between donor and acceptor sites for each FRET pair in the presence and absence of cAMP, at 0 and −100 mV. The distance dependence of our tmFRET was estimated by plotting our FRET efficiencies at 0 and −100 mV in the presence of cAMP versus the corresponding β-carbon distances (in distance/$R_0$ units) from cryo-EM structures of hHCN1 bound to cAMP (in the up and down states of the S4, respectively) (Fig. 4). This strategy is based on the assumption that spHCN in the presence of cAMP shares structural similarity

with hHCN1 based on similarities in their sequence and hyperpolarization-dependent activation[12]. For each donor site, W355Anap or C369Anap, we determined the $R_0$ using the emission spectra and quantum yields of Anap at each site at 0 and −100 mV, and the absorption spectrum of Cu$^{2+}$-TETAC (see Supplementary Fig. 5d and Methods)[15]. As previously observed[15,39], the relationship between FRET efficiency and dis-tance was somewhat shallower than predicted by the Förster equation, likely due to heterogeneity of the interatomic distances in proteins (Fig. 4b). To account for this heterogeneity, the data were fit by the convolution of the Förster equation with a Gaussian function (Förster Convolved Gaussian (FCG)) as pre-viously described[15,39]. In this case, when FRET donors were in the transmembrane segments and acceptors were in the C-linker, a Gaussian distribution of distances with a standard deviation of 5 Å could fit the distance dependence of the FRET efficiency we observed at 0 and −100 mV in the presence of cAMP (Fig. 4b). This relationship also assumes similar heterogeneity in our dif-ferent transmembrane segment-C-linker distances and provides a standard curve to convert our measured FRET efficiencies to β-carbon distances for each of the conditions in our experiment. While the distances may also reflect the rotameric states of L-Anap and Cu$^{2+}$-TETAC-modified cysteines, previous experi-ments have suggested that this effect is minimal with the short linkers of our donor and acceptor[15,39]. Figure 5 shows the dis-tances for each FRET pair in four conditions: cAMP bound at 0 mV, "resting-cAMP state"; apo at 0 mV, "resting-apo state"; cAMP bound at −100 mV, "activated state"; and apo at −100 mV, "inactivated state".

To interpret our measured distances in the context of the channel structure, we used Rosetta-based structural modeling[15,40,41]. The Rosetta software suite is a powerful set of programs for predicting the structure of proteins based on known high-resolution structures, experimental constraints, and empiri-cal rules for protein folding. It allowed us to combine known cryo-EM structures of HCN channels with our measured distances to identify rearrangements of the channel that are consistent with each of our four distances in each state. We began by generating starting models of the spHCN channel for each of the four states: resting-cAMP, resting-apo, activated, and inactivated. The starting models for resting-cAMP and resting-apo states were based on the cryo-EM structures of hHCN1 in the presence (PDB: 5U6P) and absence (PDB: 5U6O) of cAMP, respectively. The starting model for the activated state was based on the hHCN1 cyro-EM structure with its S4 trapped in the down state by a metal bridge. Because of the lack of a decrease in FRET efficiency between the S5 helix and the A′ helix with voltage steps to −100 mV in the absence of cAMP (Fig. 4d), the starting model for the inactivated state was a chimeric homology model with the amino terminus and VSD from the S4 helix-down structure (PDB code: 6UQF) and the S5–S6 and the carboxy-terminal region from the apo structure of hHCN1 (PDB code: 5U6O). We then imposed our distance measurements as constraints in the β-carbon distances in the Rosetta modeling. The modeling assumed fourfold symmetry and primarily rigid-body movements between the transmembrane domain and carboxy-terminal region (see Methods). With the above assumptions and constraints, the modeling in each state converged, with the top 17 highest scoring models for each state nearly identical (Cα RMSD <1 Å (Supplementary Fig. 7). Furthermore, the Rosetta models accurately reproduced the distances measured from our tmFRET experiments for each of the four states (Fig. 5).

**cAMP-induced rearrangement of the C-linker region of spHCN channels.** The resting-cAMP model was quite similar to

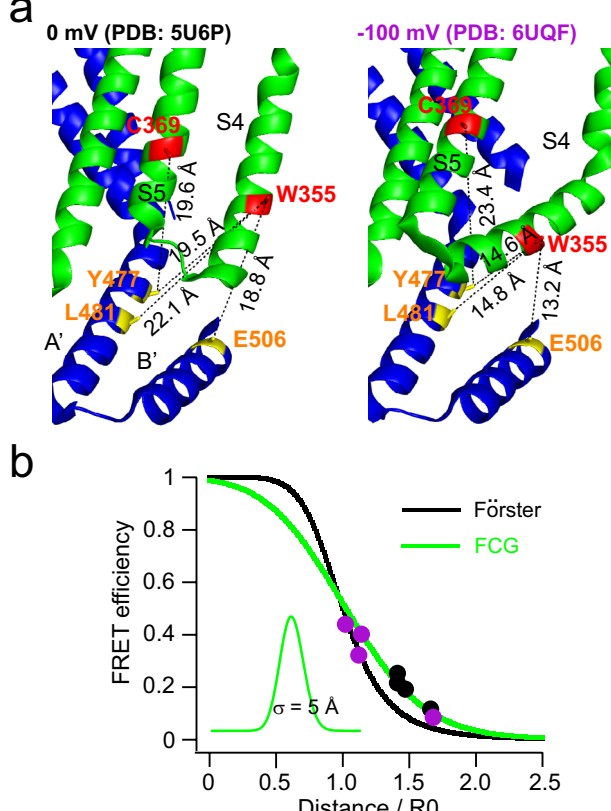

**Fig. 4 Distance determination using measured FRET efficiencies, guided by FCG and homologous structures. a** Eight FRET pairs used to obtain the Förster Convolved with Gaussian (FCG) relation to determine the distance dependence of the FRET efficiency. The Cβ–Cβ distances were measured from the homology models based on human HCN1 (PDB:5U6P and PDB: 6UQF) assuming they correspond to the structures at 0 and −100 mV respectively (in the presence of cAMP). **b** The measured FRET efficiencies are plotted versus the distances measure in panel **a** (in $R_0$ units) as 4 black (0 mV) and 4 purple (−100 mV) circles. Predicted distance dependencies of the Förster equation (black) and the FCG relation (green) are shown. The FCG relation used a Gaussian distribution with a standard deviation, σ, of 5 Å (inset).

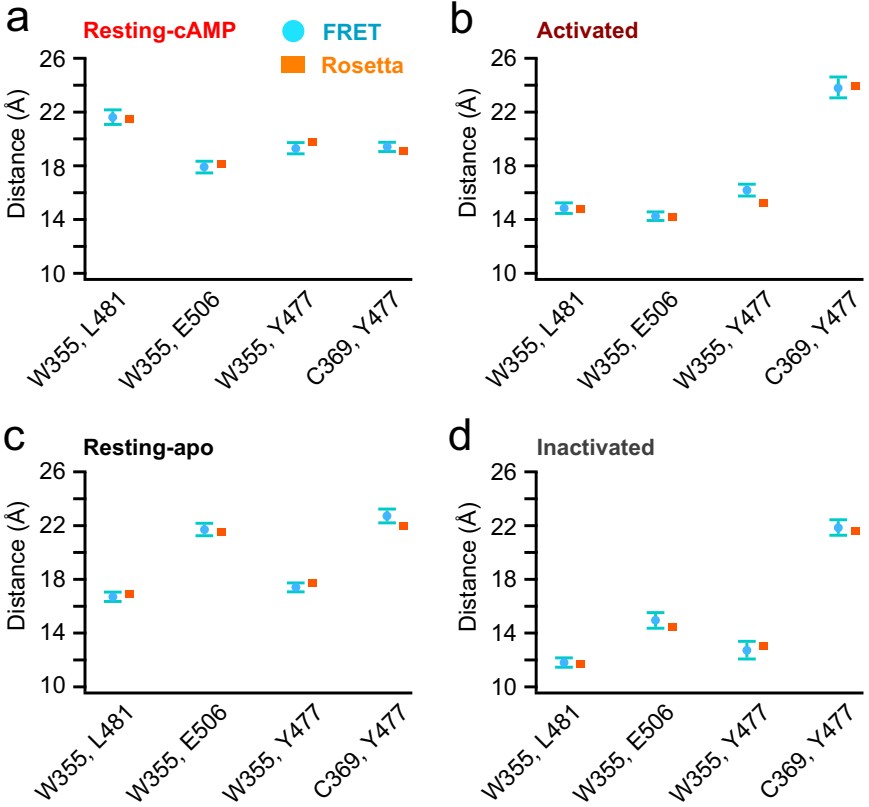

**Fig. 5 Comparison of the measured distances from tmFRET and the corresponding distances from Rosetta models. a–d** Comparison of the distances of FRET pairs measured from tmFRET (circle, mean ± s.e.m, n = 3–7) using the FCG relation in Fig. 4b and mean Cβ–Cβ distances of these FRET pairs in the 17 top scoring Rosetta models (square) based on the experimentally-determined distance constraints in four states: resting-cAMP (**a**), activated (**b**), resting-apo (**c**), and inactivated (**d**). The standard deviation of the Cβ–Cβ distances from the top 17 scoring Rosetta models for each state was less than 0.3 Å.

the cryo-EM structure of hHCN1 in the presence of cAMP[9] (RMSD = 2.0 Å), as expected from the calibration in Fig. 4. However, the resting-apo model exhibited a noticeable rearrangement of the carboxy-terminal C-linker and CNBD. These models suggest a largely rigid-body clockwise rotation of the C-linker in spHCN channel in the absence of cyclic nucleotide at 0 mV as viewed from the intracellular side (Fig. 6, Supplementary Fig. 8, and Supplementary Movie 1). This rotation causes the A′ helix to move closer to the S4 helix and further from the S5 helix, and the B′ helix to move further from the S4 helix, as predicted from the FRET measurements alone (Fig. 5a, c). In addition, in the absence of cyclic nucleotide, the C-linker moved slightly downward relative to the transmembrane domain (Fig. 6, Supplementary Fig. 8, and Supplementary Movie 1). This new position of the A′ helix in the absence of cyclic nucleotide could provide a possible link between the S4 helix and the A′ helix and might be critical for the inactivation observed in spHCN channels in the absence of cyclic nucleotide.

Comparing the inactivated to the activated models at −100 mV, the absence of cAMP in the inactivated model produced a qualitatively similar rearrangement of the C-linker as seen in the resting models, despite the S4 helix of the VSD being in the down position. Like the resting models, the C-linker undergoes a clockwise rotation around the central axis of the pore and small downward movement in the absence of cAMP (Fig. 6, Supplementary Fig. 8, and Supplementary Movie 2). The rotation, however, is somewhat smaller at −100 mV than observed at 0 mV. As discussed below, this smaller rotation appears to be due to the downward and tilted position of the carboxy-terminal region of the S4 and may underlie the inactivation of spHCN seen at −100 mV.

**Hyperpolarization-induced activation and inactivation of spHCN.** The resting-cAMP and activated models largely reflect the starting models used in the Rosetta modeling based on known HCN1 cryo-EM structures, as expected from Fig. 4. Compared to the resting-cAMP model, the activated model displays a large (~10 Å) downward movement of the S4 with a tilting motion below residue Ser[346] (Fig. 6, Supplementary Fig. 8, and Supplementary Movie 3). This rearrangement is also consistent with previous tmFRET measurements of the movement of the S4 and molecular dynamics simulations[15–17]. In addition, the activated model showed a significant bending movement of the S5, consistent with the cryo-EM structure of the hHCN1 channel with its S4 helix trapped in the downward position[16] (Fig. 6, Supplementary Fig. 8, and Supplementary Movie 3). These features were present in the starting models and were largely constrained during the modeling. Nevertheless, this movement of the S5 could almost completely account for the decrease in FRET efficiency between the S5 helix and the A′ helix with hyperpolarizing steps to −100 mV in the presence of cAMP (Fig. 3d). Like the hHCN1 structure[16], the activated model displayed little change in the relative orientation of the transmembrane domain and C-linker relative to the resting-cAMP model (Supplementary Movie 3). Indeed, the activated spHCN model and the hHCN1 model with the S4 helix trapped in the down position had an RMSD of only 1.91 Å. These results suggest that the hHCN1 S4-down structure[16] also well represents the spHCN activated structure (at −100 mV in the presence of cAMP), and the mechanism of hyperpolarization-dependent activation of spHCN channels is likely to be similar to that of mammalian HCN channels.

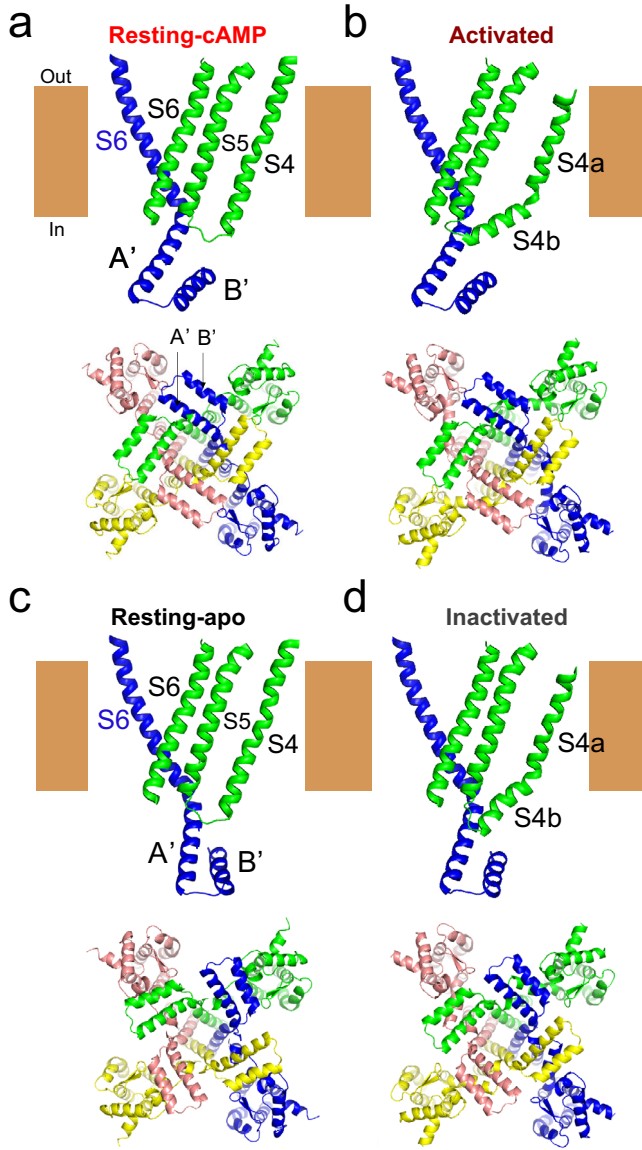

**Fig. 6 Rosetta models for activation and inactivation of spHCN channels and their regulation by cAMP. a–d** Modeled structures (side view and bottom view) highlighting the rearrangement of the C-linker region of one subunit relative to the S4 and S5 helices of the adjacent subunit for four states: resting-cAMP (**a**), activated (**b**), resting-apo (**c**), and inactivated (**d**). The S4 helix moves downward and bends into two segments after hyperpolarization in both activated and inactivated states. The bottom view highlights the rigid-body rotational movements of the C-linker around the central axis of the pore in the absence of cyclic nucleotide. See also Supplementary Movies 1–4.

The conformational rearrangements between the resting-apo and inactivated models showed features distinct from those in the presence of cAMP (Fig. 6, Supplementary Fig. 8, and Supplementary Movie 4). Unlike the models in the presence of cAMP, the inactivated model displayed a considerable rotation of the C-linker relative to the resting-apo model (compare Supplementary Movies 3 and 4). The rotation appears to result from an interaction between the carboxy-terminal region of the S4 helix and the A′ helix, causing the downward and tilting motion of the S4 to be translated into a rotation of the C-linker (Supplementary Movie 4). The counterclockwise rotation of the C-linker at −100 mV in the absence of cyclic nucleotide is in the direction expected

to wind the S6 bundle and might be responsible, at least in part, for pore closure (inactivation). These models, therefore, provide a potential molecular mechanism for how hyperpolarization causes activation in the presence of cAMP and inactivation in the absence of cAMP for spHCN channels.

## Discussion

All HCN channels are activated by membrane hyperpolarization. However, unlike mammalian HCN channels, spHCN channels also undergo hyperpolarization-dependent inactivation in the absence of cyclic nucleotide which is eliminated in the presence of cAMP. Using tmFRET, we show that the movement of the S4 is similar in the presence and absence of cAMP, suggesting that the inactivation does not result from a change in the S4 movement, but from a conformational change that is coupled to the S4 movement, exclusively in the absence of cAMP. Furthermore, we show that, in the absence of cyclic nucleotide, there is a substantial repositioning of the A′ helix of the C-linker region closer to the S4 segment. Using Rosetta modeling of our tmFRET results, we show how the close proximity between the S4 helix and the A′ helix might allow the downward movement of the S4 helix with hyperpolarization to be coupled to the pore via the A′ helix (Fig. 6 and Supplementary Movie 4). This coupling could make opening of the channel less favorable, leading to inactivation. We therefore propose that the A′ helix may mediate the coupling between the S4 segment and the PD during voltage-dependent inactivation.

This hypothesis for inactivation is consistent with a number of previous observations in CNBD-family channels. It has previously been shown that, unlike the N-type or C-type inactivation in voltage-dependent potassium (Kv) channels[42], the inactivation in spHCN channels involves the same gate as activation, the bundle crossing at the bottom of the S6[13]. This idea of using the same gate for both activation and inactivation of spHCN is compatible with our proposed mechanism for inactivation. While the mechanism for inactivation was originally proposed to involve a loss of coupling between the VSD and the pore (also referred to as desensitization to voltage or "slippage")[13], that mechanism cannot account for the constitutive opening produced by many mutations that disrupt electromechanical coupling of spHCN channels[14]. Our Rosetta model suggests that a separate new coupling pathway contributes to the inactivation rather than a simple loss of coupling for activation. Perhaps most importantly, in spHCN channels, deletion of the carboxy-terminal domain, including the A′ helix, eliminates the inactivation process while leaving the hyperpolarization-dependent activation intact[32,33]. Similarly, removing the S4–S5 linker, which generates a split channel with a break in the backbone between the VSD and the PD, attenuates the inactivation[14]. In addition, cross-linking the A′ helix to the S4–S5 linker with a $Cd^{2+}$ bridge can produce channels that are closed at hyperpolarized voltages but open at depolarized voltage (depolarization-activated)[43]. All of these observations are consistent with the A′ helix mediating the coupling between the S4 segment and the pore during voltage-dependent inactivation.

The C-linker, including the A′ helix, is common to all CNBD family channels[6]. The A′ helix has been shown to rearrange after cyclic-nucleotide binding to promote the pore opening, which was further corroborated in the new cryo-EM structures of a eukaryotic CNG channel TAX-4 and a human rod photoreceptor CNGA1 channel[11,30,44–47]. Our models display an upward movement of the C-linker after cyclic nucleotide binding similar to other CNBD family channels[44,46,48,49]. However, a counterclockwise rotation of the A′–B′ helix around the central axis of the pore after cAMP binding (Supplementary Movies 1 and 2)

was also revealed in our tmFRET measurements and modeling. This movement has not been observed in other CNBD family channels, including the rearrangement between the open and closed conformations of TAX4[44] or human CNGA1[47], and may underlie why this type of inactivation is only observed in spHCN channels but not in other HCN or CNG channels.

The hyperpolarization-dependent activation of HCN channels involves a different coupling pathway, independent of the A′ helix or the short S4–S5 linker. HCN channels with a deletion of either the A′ helix or the S4–S5 linker maintain their hyperpolarization-dependent activation[14,32,33,50]. Instead, mutations of the S4 and S5 segments can eliminate the hyperpolarization-dependent activation and uncover depolarization-dependent activation[14,21,22]. This depolarization-dependent activation is eliminated by the addition of cAMP, reminiscent of the hyperpolarization-dependent inactivation in spHCN channels[14]. Collectively, it appears that there are two separate coupling pathways for voltage-dependent regulation of spHCN channels. The hyperpolarization-dependent activation seems to involve a stabilization of the closed conformation of the pore by the "up state" of the S4 helix via the S5 helix, while the hyperpolarization-dependent inactivation seems to involve stabilization of the closed (inactivated) state via the A′ helix. A mixture of these two pathways could explain the "bell-shaped" and "U-shaped" G-V relationships often observed in the CNBD-family channels[14,22,51].

How might the position of the A′ helix in the absence of cyclic nucleotide couple the downward movement of the S4 to channel closing (inactivation)? Morphs between our proposed resting-apo and inactivated structures suggest that hyperpolarizing voltage steps elicit a counterclockwise rotation of the A′ helix of the C-linker around the pore central axis (Supplementary Movie 4) which could twist the bundle crossing closed. In contrast, in the presence of cAMP, there is no rotation (Supplementary Movie 3). The inactivation, therefore, might result from an interaction between the S4 or S4–S5 linker and the A′ helix in the absence of cyclic nucleotide so that the tilting motion of the S4 essentially drags the A′ helix in a counterclockwise direction (Supplementary Movie 4). In the presence of cAMP, the S4 is further from the A′ helix and might interact more weakly or differently with the A′ helix. In this sense, the A′ helix seems to function as a clutch for inactivation. In the absence of cyclic nucleotide, the clutch is engaged allowing the downward movement of the S4 to close (inactivate) the pore. Like pushing the clutch pedal on a car, cAMP disengages the clutch and the downward movement of the S4 no longer causes channel closure. For mammalian HCN channels, it is likely that this A′ helix clutch is mostly disengaged so that they do not inactivate even in the absence of cyclic nucleotide. We speculate that the A′ helix of mammalian channels might rotate differently in the absence of cAMP, so that it is not positioned to couple the movement of the S4 to channel inactivation. In general, coupling is an inherently energetic phenomenon, and it is currently very difficult to estimate energetics from structures alone.

The mechanism we propose might be relevant to other channels in the CNBD family. KCNH channels have an overall architecture similar to HCN channels, including a non-domain-swapped VSD, a C-linker, and a cyclic nucleotide-binding homology domain (CNBHD)[52]. However, KCNH channels are activated by membrane depolarization instead of by membrane hyperpolarization. It is worth noting that the hyperpolarization-dependent inactivation in spHCN can also be viewed as a depolarization-dependent activation. Indeed, we found the C-linker movement of spHCN in the absence of cAMP in response to depolarizing voltage steps resembles that postulated for KCNH channels by morphing the closed EAG structure and the open hERG structure[53–55] (Supplementary Movie 5). We postulate that the A′ helix may function similarly for the depolarization-dependent activation of KCNH channels and the hyperpolarization-dependent inactivation of spHCN channels. This A′ helix-mediated pathway has also been proposed to underlie the calmodulin modulation and "Cole-Moore" effect of KCNH (EAG) channels[55]. Furthermore, a role for the A′ helix in hyperpolarization-dependent activation has also been proposed for KAT1 channels, a plant CNBD channel[26]. Interestingly, in the canonical domain-swapped VGICs, the S4–S5 linker forms a long helix that is in a structurally-similar position to the A′ helix of CNBD channels, and has been proposed to have a similar functional role, to couple the downward movement of the S4 to closure of the channel[20,23,56].

In conclusion, the CNBD family channels could have evolved two noncanonical coupling pathways between the VSD and the PD: hyperpolarization-dependent stabilization of the open state that involves the amino-terminal part of S5 helix and depolarization-dependent stabilization of the open state that involves the A′ helix of the C-linker. These two pathways for VSD-PD coupling might act nearly simultaneously to affect the overall energetics of channel opening and can explain the diverse voltage-dependent gating behaviors of the CNBD family of channels.

## Methods

**Molecular biology.** The full-length spHCN cDNA (a gift from U. B. Kaupp, Molecular Sensory Systems, Center of Advanced European Studies and Research, Bonn, Germany; GenBank: Y16880) was subcloned into a modified pcDNA3.1 vector (Invitrogen, Carlsbad, CA) that contained a carboxy-terminal eYFP, a T7 promoter, and 5′- and 3′-untranslated regions of a *Xenopus* β-globin gene. For the ACCuRET experiments, the background cysteine-depleted spHCN construct contained mutations: C211A, C224A, C369A, and C373A. Quickchange II XL Site-Directed Mutagenesis kit (Agilent technologies, Santa Clara, CA) or overlapping PCR was used for mutagenesis. Fluorescence-based DNA sequencing was carried out by Genewiz LLC, Seattle, WA. All oligo primers used for mutagenesis, including the names and sequences, are listed in the Supplementary Table 2. The in vitro mRNA synthesis was done using the HiScribe T7 ARCA mRNA Kit (New England Biolabs, Ipswich, MA) or the mMESSAGE T7 ULTRA Transcription Kit (ThermoFisher, Waltham, MA).

**Oocyte expression and electrophysiology.** The animal-use protocols were consistent with the recommendations of the American Veterinary Medical Association and were approved by the Institutional Animal Care and Use Committee (IACUC) of the University of Washington. L-Anap (free-acid form, AsisChem, Waltham, MA) was made as a 1 mM stock in water at a high pH by adding NaOH as previously published[15]. The pANAP plasmid (purchased from Addgene, Cambridge, MA) contained the orthogonal tRNA/aminoacyl-tRNA synthetase specific to L-Anap[57]. Inside-out configuration of the patch-clamp technique was implemented with an EPC-10 (HEKA Elektronik, Germany) or Axopatch 200B (Axon Instruments, Union City, CA) patch-clamp amplifier and PATCHMASTER software (HEKA Elektronik)[15]. Solution changing was done using a μFlow microvolume perfusion system (ALA Scientific Instruments, Farmingdale, NY). For oocyte patch-clamp recording and Co$^{2+}$-HH tmFRET experiments, the standard bath and pipette saline solutions contained 130 mM KCl, 10 mM HEPES, 0.2 mM EDTA (pH 7.2). 1 mM CoSO$_4$ was added to the perfusion solution with EDTA eliminated. For ACCuRET experiments, stabilization buffer (SBT) contained 130 mM KCl, 30 mM TRIS, 0.2 mM EDTA (pH 7.4) and was used for both the pipette and the bath solution. A +40 mV voltage was used to facilitate tail-current measurements. The Cu$^{2+}$-TETAC (used at a final concentration of 20 μM) was prepared as previously described[39].

**Fluorescence measurements.** PCF and spectral measurements were done similar to previously published[15]. Our PCF experiments used a Nikon Eclipse TE2000-E inverted microscope with a 60 × 1.2 NA water immersion objective. Epifluorescence recording of L-Anap was performed with wide-field excitation using a Lambda LS Xenon Arc lamp (Sutter Instruments) and a filter cube containing a 376/30 nm excitation filter and a 460/50 nm emission filter. A 425 nm long-pass emission filter was used for the spectral measurement of L-Anap. YFP was measured with a filter cube containing a 490/10 nm excitation filter and a 535/30 nm emission filter. Images were collected with a 200 ms exposure using an Evolve 512 EMCCD camera (Photometrics, Tucson, AZ) and MetaMorph software (Molecular Devices, Sunnyvale, CA). For spectral measurements, images were collected by a spectrograph (Model: 2150i, 300 g/mm grating, blaze = 500 nm; Acton research,

Acton, MA) mounted between the Nikon microscope and the Evolve 512 EMCCD camera.

**Data analysis**. Data were analyzed using IgorPro (Wavemetrics) and Image J (NIH), similar to previously published[15].

The conductance-voltage (G-V) relationships were measured from the instantaneous tail currents at +40 mV following voltage pulses from 0 mV to between 0 and −110 mV. The relative conductance was plotted as a function of the voltage of the main pulse and fitted with a Boltzmann Eq. (1):

$$G/G_{max} = 1/(1 + \exp[(V - V_{1/2})/V_s])  \quad (1)$$

where $V$ is the membrane potential, $V_{1/2}$ is the potential for half-maximal activation, and $V_s$ is the slope factor.

Fluorescence images from the membrane patches were imported into ImageJ[58] for analysis. Regions of interest were determined manually around the dome of the patch, excluding any regions of the glass of the pipette (Supplementary Fig. 3b, d). For each patch, a background subtraction was performed by selecting an area outside of the patch inside the pipette. The mean subtracted fluorescence intensity ($fl$) was used for the following calculations.

The FRET efficiency was calculated using the following Eq. (2) as previously described[34]:

$$FRET_{eff} = \frac{F_{no\,HH} - F_{HH}}{F_{HH} * F_{no\,HH} + F_{no\,HH} - F_{HH}} \quad (2)$$

$$\text{or } FRET_{eff} = \frac{F_{no\,Cys} - F_{Cys}}{F_{Cys} * F_{no\,Cys} + F_{no\,Cys} - F_{Cys}}$$

Where $F_{HH}$ and $F_{no\,HH}$ are the fractions of fluorescence that are unquenched by $Co^{2+}$ in channels with and without HH sites, respectively; $F = \frac{fl\,(metal)}{fl\,(no\,metal)}$. Similarly, $F_{Cys}$ and $F_{no\,Cys}$ are the fractions of fluorescence that are unquenched by $Cu^{2+}$-TETAC in channels with and without cysteines, respectively.

The $R_0$ values, the distances that predict 50% energy transfer, were calculated using the established Eq. (3)[39].

$$R_0 = C\sqrt[6]{JQ\eta^{-4}\kappa^2} \quad (3)$$

Where $C$ is the scaling factor, $J$ is the overlap integral of the donor emission spectrum and the acceptor absorption spectrum, $Q$ is the quantum yield of the donor, $\eta$ is the index of refraction, and $\kappa^2$ is the orientation factor. $\eta$ was assumed to be 1.33, and $\kappa^2$ was assumed to be 2/3, consistent with what was previously reasoned and used[15]. The $J$ values for each L-Anap site at 0 and −100 mV in the presence and absence of cAMP were calculated using the emission spectrum of each site at each voltage and the absorption spectrum of $Cu^{2+}$-TETAC in solution. The relative quantum yields of L-Anap at each site at 0 and −100 mV were estimated from the slopes of the plots of Anap intensity vs. YFP intensity at 0 mV and the relative brightness at −100 mV vs. 0 mV ($fl_{Anap,\,-100\,mV} / fl_{Anap,\,0\,mV}$) as previously published (Supplementary Fig. 5)[15]. $R_0$ values calculated using the estimated spectral overlaps and quantum yields were as follows: 13.3 Å (0 mV) and 13.0 Å (−100 mV) for W355Anap both in the presence and absence of cAMP; and 13.9 Å (0 mV) and 13.9 Å (−100 mV) for C369Anap both in the presence and absence of cAMP. No spectral shifts due to hyperpolarization to −100 mV were observed for either W355Anap or C369Anap emission.

**Rosetta modeling**. Homology models of the spHCN channel based on the human HCN1 structure[9] were created using RosettaCM without any experimental constraints[40,41]. PDB codes used for the four starting models are: 5U6P for the "resting-cAMP" (cAMP, 0 mV) model, 5U6O for the "resting-apo" (no cyclic nucleotide, 0 mV) model, 6UQF for the "activated" (cAMP, −100 mV) model, and a chimeric combination of 6UQF (amino-terminal end to transmembrane S1–S4) and 5U6O (S5 helix to the carboxy-terminal end) for the "inactivated" (no cyclic nucleotide, −100 mV) model. Four spHCN channel models: resting-cAMP, resting-apo, activated, and inactivated were built and refined using RosettaCM similar to previously published[15].

The structures were refined using RosettaCM with a combination of three types of restraints: (1) coordinate restraints on every non-hydrogen backbone atom tethering the absolute Cartesian coordinates; (2) atom-pair constraints to all backbone atom pairs within some cutoff distance tethering atoms in close proximity to maintain their distance; and (3) reference constraints using experimentally-derived distances based on our tmFRET measurements. Coordinate constraints were applied to the amino-terminal region and S1–S4 transmembrane domain (amino acids 167–362). Atom-pair constraints were applied to the S5–S6 transmembrane domain (amino acids 363–511) without symmetry and the carboxy-terminal domain (amino acids 515–657) with symmetry. This was done to "rigidify" the structure while still allowing for domain movements. All three types of constraints used a harmonic potential.

During refinement, there were four key parameters that controlled refinement behavior: three weights controlling the relative weigh on each of the three constraint types, and the distance cutoff used for the atom-pair constraints. A 4D grid search was carried out where we considered: weights of 1, 2, and 4 kcal/mol/Å² for the atom-pair and coordinate restraints; weights of 10, 20, 30, and 40 kcal/mol/

Å² for reference model restraints; and 6, 8, and 10 Å for our atom-pair distance cutoff. We then chose the refined structure with strongest "reference model" restraints and weakest experimental restraints that satisfied measured distances to within 1 Å in all four models. This resulting structure had a coordinate constraint weight of 1 kcal/mol/Å², an atom-pair constraint weight of 2 kcal/mol/Å², an atom-pair-constraint cutoff distance of 6 Å, and experimental constraint weight of 40 kcal/mol/Å². The resulting models were the ones that minimally perturbed local structure while satisfying the experimental data. Structural representations and morphs were created using the PyMOL software (https://pymol.org).

**Statistics and Reproducibility**. Data parameters were expressed as mean ± s.e.m. of $n$ independent patches. Statistical significance (*$p < 0.05$; **$p < 0.001$) was determined using a one-way ANOVA and Tukey's post hoc test. The mean and s.e. m. for the FRET efficiency of TETAC experiments were calculated using the fractional quenching measurements (i.e., $F_{Cys}$ and $F_{no\,Cys}$) in Monte Carlo resampling (NIST Uncertainty Machine v1.3.4; https://uncertainty.nist.gov) as previously published[15]. For their individual data points and statistical tests, FRET efficiencies were calculated with Eq. (2), using $F_{Cys}$ from independent experiments and the averaged $F_{no\,Cys}$ from all experiments under the same condition.

**Reporting Summary**. Further information on research design is available in the Nature Research Reporting Summary linked to this article.

## Data availability
Data supporting the findings of this manuscript are available from the corresponding authors upon reasonable request. A reporting summary for this Article is available as a Supplementary Information file. Source data are provided with this paper.

## Code availability
Rosetta models and Rosetta scripts were uploaded to GitHub: https://github.com/zagotta/NCOMMS2021.

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

## Acknowledgements

We thank all members of the Zagotta laboratory for their advice and support. This work was funded by NIH Grants R01EY010329, R01GM125351, and R01GM127325 (to W.N.Z.), and F32NS077622 (to T.K.A.).

## Author contributions

G.D., T.K.A. and W.N.Z designed the experiments. G.D. performed the experiments. T.K.A performed pilot experiments. F.D.M., G.D. and W.N.Z. designed Rosetta modeling. F.D. performed Rosetta modeling and wrote the Rosetta modeling methods of the manuscript. G.D., T.K.A, F.D. and W.N.Z. analyzed the data. G.D. and W.N.Z. wrote the manuscript.

## Competing interests

The authors declare no competing interests.
