## [Peer Review File · Nature Communications]

Reviewer #1 (Remarks to the Author):

In this manuscript, the authors used a combination of patch fluorometry, tmFRET and Rosetta modeling to reveal the conformational changes associated with hyperpolarization and cAMP induced spHCN channel activation and inactivation. They observed relative movements between the transmembrane helices and intracellular domains, which suggested a novel electromechanical coupling mechanism in the CNBD channel family. This is an important and interesting study, as it provides crucial information regarding changes in distances between channel domains while the channels are executing their physiological functions in their native lipid environment. To improve this manuscript, there are some concerns to be addressed:

Major concerns:

1. The authors quantified the FRET efficiency based on the fluorescence intensity of ANAP. As the emission spectrum of ANAP is environmentally sensitive, will the conformational rearrangements in S4 lead to shifts in ANAP emission spectrum when ANAP was incorporated at S346 and W355? How would shifts in ANAP emission (if there is any) impact the accuracy of calculation of FRET efficiency?
2. When the authors measured FRET between ANAP incorporated at W355 in S4 and intracellular domains, will the fluorescence of ANAP itself be altered when S4 moves during hyperpolarization? The authors provided a solid control experiments on ANAP incorporated at C369 in S5 in Fig. S5c, demonstrating that ANAP fluorescence itself is not affected by hyperpolarization, so that the FRET changes they observed between C369ANAP and intracellular sites are truly due to the distance changes. Therefore, the author should also perform negative control experiments like Fig. S5c in which ANAP fluorescence measured from W355ANAP only channels, which would show that the FRET changes they saw between W355ANAP and intracellular sites are not affected by any variations in ANAP fluorescence itself.
3. The interpretation of changes in FRET efficiencies relies heavily on the accuracy of Rosetta modeling. In Methods, the authors should specify how the distance information from FRET experiments is translated into constraints. For “(3) reference constraints using experimentally-derived distances based on our tmFRET measurements”, I would assume atom-pair constraints are imposed, if so, are the constraints applied to the distance between C beta atoms of residues? We know that though the tm metal ions do not assume a specific orientation, the sidechain rotamers of ANAP and TETAC may take certain preferred conformation. As the authors used Rosetta to do the modeling, it is suggested that they generate the rotamer libraries of ANAP and TETAC, and then use Rosetta to calculate the preferred sidechain configuration of ANAP and TETAC when the backbones of these labeling sites (S4 and intracellular domains) are in different conformational states. My concern is, for instance, for ANAP incorporated at W355 site, the sidechain of ANAP takes one preferred conformation in the resting state and another conformation in the activated and/or inactivated state. As the interpretation of FRET changes is about distances changes in a few angstroms, the sidechain configuration of ANAP may impact considerably as the sidechain itself is relatively large.
4. As the Rosetta modeling relies on Monte Carlo simulation, which is probabilistic. How were the models presented in this manuscript converged during modeling? The authors should specify how many models were generated for modeling of each state and how the final models were chosen in Methods. If the models presented are the top scored models in each RosettaCM trials, how large would the rmsd be between top ten models in each trial? For instance, in Fig. 5 the variation in FRET

measurements was shown, such variation in Rosetta modeling as reflected by rmsd between top models should also be estimated and compared.

Minor point:

On page 4, “($\Delta F/F = 46.3 \pm 2.2\%$ in cGMP and $\Delta F/F = 26.0 \pm 3.3\%$ in the absence of cyclic nucleotide) (Fig. 2b)” Fig. 2b should be Fig. S2b.

Reviewer #2 (Remarks to the Author):

In the manuscript “NONCANONICAL ELECTROMECHANICAL COUPLING MECHANISM OF AN HCN CHANNEL” Dai et al investigate the interesting and non-trivial question about how voltage and ligand binding are used together to modulate hyperpolarization activated channels (HCN). They use HCN from sea urchin (spHCN) as model to understand movements of the voltage sensing S4 helix and the C-linker in the absence and presence of the agonist, cAMP, by means of electrophysiology and transition metal FRET. Distance constraints obtained from these experiments are used to model conformational changes in different states of the channel (resting, activated, inactivated) using Rosetta and structures of human HCN1. spHCN have an intriguing property that differs from other HCN channels, inactivation in the absence of ligand which disappears in its presence, which always puzzled researchers, and this manuscript does a great job exploring it and proposing a mechanism.

The work is beautiful, convincing, and scientifically sound, the experiments are logical and well thought-out, controls are in place, the manuscript is overall well-written, although some sections would benefit from further clarification.

Issues for revision:

1. The abstract could be streamlined. The part describing the findings of the paper is confusing, not sufficiently descriptive to understand the message. The clutch concept is not helpful here, probably because there is too little space for it to be properly explained (it is helpful later on, in the discussion). Too much undefined jargon in the abstract too (Aprime helix, S5, CNBD).
2. Introduction, page 1, define hydrophobic constriction site.
3. Also in the introduction (page 3), they describe cAMP as full agonist of most HCN channels. HCN channels are activated by hyperpolarization and the activity is only modulated by cAMP, which, per definition, is not a full agonist of HCN channels. Furthermore, cAMP displays significantly different efficacies depending on the isoform of HCN. Most importantly here, cAMP shows almost no effect with HCN1, which the authors later use as their starting point for all Rosetta modeling. The authors should be more precise in their description of cAMP efficacies.
4. End of intro. The proposed mechanism is unclear (again, the clutch concept does not come through here). Reconsider rewriting the last 2 sentences.
5. In order to measure tail currents, voltage steps to +40 mV are performed (and no activity is observed). What is the reason for stepping to +40 mV instead of 0 mV, which the authors use as the holding potential?
6. The structural conformations with respect to the C-linker as discussed in Figure 6 are hard to appreciate in the side-by-side representation. The authors could consider adding a third column and show overlays of one subunit in the respective states to visually relay the differences and their interpretation more clearly.
7. On page 9 and 11, the interpretation that the CNBD undergoes a counterclockwise rotation upon

cAMP binding is interesting. Can the authors be more specific about how this compares with what was seen in TAX4 upon cGMP binding? Is it the same direction of rotation or opposite? How do the magnitudes of rotation compare? The authors need to provide some rationale as to why spHCN may be different. A figure would also be useful with overlays between spHCN and TAX4 apo to agonist-bound (resting).

8. In light of that, the authors should present measurements, as shown in Figure 4, also for the structural models obtained from their Rosetta modeling. Is it possible that in the activated/inactivated states the signal from different subunits contribute predominantly to tmFRET? In other words, how do the authors make sure, that the observed tmFRET signal reports on the same donor/acceptor pair in all functional states?

9. The differences with HCN1 are really a most intriguing part of this manuscript. Discussion page 12, the authors state that “this clutch might be always disengaged so that HCN1 channels do not inactivate”. Can you provide a framework for this statement? How do you envision the clutch being disengaged in a structural way?

Minor points:

The title could be improved. It says very little. “Noncanonical” is vague and not particularly appropriate here, since there is no “canonical” mechanism for spHCN inactivation

Page 4, 6th line from bottom, should be Fig. S2b instead of Fig.2b

Page 6, clarify if the cysteine-reduced spHCN works like WT.

On page 7, second to last paragraph: “There results suggest...” probably should say These results suggest...

On page 11, last paragraph: “The A’ helix have been shown...” probably should be The A’ helix has been shown...

In all supplementary figures the respective voltages (holding, hyperpolarization, tail) should be indicated in the panels.

REVIEWER COMMENTS

Reviewer #1 (Remarks to the Author):

In this manuscript, the authors used a combination of patch fluorometry, tmFRET and Rosetta modeling to reveal the conformational changes associated with hyperpolarization and cAMP induced spHCN channel activation and inactivation. They observed relative movements between the transmembrane helices and intracellular domains, which suggested a novel electromechanical coupling mechanism in the CNBD channel family. This is an important and interesting study, as it provides crucial information regarding changes in distances between channel domains while the channels are executing their physiological functions in their native lipid environment. To improve this manuscript, there are some concerns to be addressed:

Major concerns:

1. The authors quantified the FRET efficiency based on the fluorescence intensity of ANAP. As the emission spectrum of ANAP is environmentally sensitive, will the conformational rearrangements in S4 lead to shifts in ANAP emission spectrum when ANAP was incorporated at S346 and W355? How would shifts in ANAP emission (if there is any) impact the accuracy of calculation of FRET efficiency?

For spHCN-W355Anap, there is no shift in Anap emission spectrum, which was published in our previous paper (Nat Struct Mol Biol. 2019 Aug;26(8):686-694). Also, in this previous work, S346Anap showed a shift in Anap emission, but the S346Anap was not used to determine distance in this paper. In any case, the shift in Anap emission does not impact the FRET efficiency measurement because the efficiency is determined separately for each state. For example, we used only the Cu-TETAC induced quenching of Anap at -100 mV to determine the FRET efficiency at -100 mV, not the changes in fluorescence with steps from 0 to -100 mV. We have clarified this in the manuscript and referenced the previous paper.

2. When the authors measured FRET between ANAP incorporated at W355 in S4 and intracellular domains, will the fluorescence of ANAP itself be altered when S4 moves during hyperpolarization? The authors provided a solid control experiments on ANAP incorporated at C369 in S5 in Fig. S5c, demonstrating that ANAP fluorescence itself is not affected by hyperpolarization, so that the FRET changes they observed between C369ANAP and intracellular sites are truly due to the distance changes. Therefore, the author should also perform negative control experiments like Fig. S5c in which ANAP fluorescence measured from W355ANAP only channels, which would show that the FRET changes they saw between W355ANAP and intracellular sites are not affected by any variations in ANAP fluorescence itself.

In contrast to C369Anap, there is a small decrease (~15%) in the fluorescence intensity (due to an environmentally-sensitive decrease in Anap quantum yield) of W355Anap by -100 mV hyperpolarization. This environmental sensitivity of W355Anap due to voltage was published in our previous paper (Nat Struct Mol Biol. 2019 Aug;26(8):686-694). As discussed above, this change does not impact the FRET efficiency measurement because the efficiency is determined separately for each state. In addition, the

R0 at -100 mV was corrected for this decrease in quantum yield. We have clarified this in the manuscript and referenced the previous paper.

3. The interpretation of changes in FRET efficiencies relies heavily on the accuracy of Rosetta modeling. In Methods, the authors should specify how the distance information from FRET experiments is translated into constraints. For “(3) reference constraints using experimentally-derived distances based on our tmFRET measurements”, I would assume atom-pair constraints are imposed, if so, are the constraints applied to the distance between C beta atoms of residues? We know that though the tm metal ions do not assume a specific orientation, the sidechain rotamers of ANAP and TETAC may take certain preferred conformation. As the authors used Rosetta to do the modeling, it is suggested that they generate the rotamer libraries of ANAP and TETAC, and then use Rosetta to calculate the preferred sidechain configuration of ANAP and TETAC when the backbones of these labeling sites (S4 and intracellular domains) are in different conformational states.

My concern is, for instance, for ANAP incorporated at W355 site, the sidechain of ANAP takes one preferred conformation in the resting state and another conformation in the activated and/or inactivated state. As the interpretation of FRET changes is about distances changes in a few angstroms, the sidechain configuration of ANAP may impact considerably as the sidechain itself is relatively large.

We used pairwise constraints on β -carbon distances, which is now explicitly stated, and have added the RosettaCM scripts to the supplementary information. We agree with the reviewer that there is uncertainty in how well the FRET measurements reflect the β -carbon distances, due to rotameric states of Anap and TETAC, a problem that is present in all studies using structural probes. Unfortunately, it is not yet possible to definitely calculate what the rotameric distribution is in the context of modeled structures and in native membranes. However, we believe this issue is minimal for the following reasons: (1) We utilized probes with as short a linker as possible, including using a noncanonical amino acid fluorophore with only one atom between the β -carbon and the fluorophore. (2) Our measurements of distances at 0 and -100 mV in the presence of cAMP were consistent with the cryo-EM structures of HCN1. (3) Our distances could all be accounted for by a nearly rigid body movement of the carboxy-terminal domain and the transmembrane domain, providing confidence that the distance measurements came from backbone changes, not just from a rotameric change at a particular site. And (4) we accounted for the rotameric states of our donor and acceptor by assuming the distance between the donor and acceptor has a Gaussian distribution, as illustrated in the Förster Convolved Gaussian (FCG) strategy in Fig 4. Nevertheless, we added in the text that there could still be a small effect of the rotameric distribution to our distance estimates.

4. As the Rosetta modeling relies on Monte Carlo simulation, which is probabilistic. How were the models presented in this manuscript converged during modeling? The authors should specify how many models were generated for modeling of each state and how the final models were chosen in Methods. If the models presented are the top scored models in each RosettaCM trials, how large would the rmsd be between top ten models in each trial? For instance, in Fig. 5 the variation in FRET measurements was shown, such variation in Rosetta modeling as reflected by rmsd between top models should also be estimated and compared.

As suggested, we have added a new supplementary figure (Fig S7) showing an overlay of the 17 lowest energy models using our RosettaCM methods. These models have an RMSD of less than 1 Å for each state and a standard deviation for the C β -C β distances measured for each of the FRET pairs of less than 0.3 Å which indicates convergence of the models. We have added this information to the text and figure legend.

Minor point:

On page 4, “($\Delta F/F = 46.3 \pm 2.2\%$ in cGMP and $\Delta F/F = 26.0 \pm 3.3\%$ in the absence of cyclic nucleotide) (Fig. 2b)” Fig. 2b should be Fig. S2b.

Fixed. Thanks!

Reviewer #2 (Remarks to the Author):

In the manuscript “NONCANONICAL ELECTROMECHANICAL COUPLING MECHANISM OF AN HCN CHANNEL” Dai et al investigate the interesting and non-trivial question about how voltage and ligand binding are used together to modulate hyperpolarization activated channels (HCN). They use HCN from sea urchin (spHCN) as model to understand movements of the voltage sensing S4 helix and the C-linker in the absence and presence of the agonist, cAMP, by means of electrophysiology and transition metal FRET. Distance constraints obtained from these experiments are used to model conformational changes in different states of the channel (resting, activated, inactivated) using Rosetta and structures of human HCN1. spHCN have an intriguing property that differs from other HCN channels, inactivation in the absence of ligand which disappears in its presence, which always puzzled researchers, and this manuscript does a great job exploring it and proposing a mechanism.

The work is beautiful, convincing, and scientifically sound, the experiments are logical and well thought-out, controls are in place, the manuscript is overall well-written, although some sections would benefit from further clarification.

Issues for revision:

1. The abstract could be streamlined. The part describing the findings of the paper is confusing, not sufficiently descriptive to understand the message. The clutch concept is not helpful here, probably because there is too little space for it to be properly explained (it is helpful later on, in the discussion). Too much undefined jargon in the abstract too (Aprime helix, S5, CNBD).

As suggested, we have removed the clutch idea from the abstract, making it more streamlined. We replaced with more explanation to clarify the message. In addition, we have removed undefined jargon as much as possible.

2. Introduction, page 1, define hydrophobic constriction site.

As suggested, we have added the definition of HCS.

3. Also in the introduction (page 3), they describe cAMP as full agonist of most HCN channels. HCN channels are activated by hyperpolarization and the activity is only modulated by cAMP, which, per

definition, is not a full agonist of HCN channels. Furthermore, cAMP displays significantly different efficacies depending on the isoform of HCN. Most importantly here, cAMP shows almost no effect with HCN1, which the authors later use as their starting point for all Rosetta modeling. The authors should be more precise in their description of cAMP efficacies.

Thanks for pointing this out. cAMP has been shown to be a full agonist for the CNBD, but not for the pore. We replaced “full agonist” with “full agonist for the CNBD” and referenced our previous paper using DEER studying the effects of different cyclic nucleotides on the conformational change in the CNBD (DeBerg et al., JBC, 2016).

4. End of intro. The proposed mechanism is unclear (again, the clutch concept does not come through here). Reconsider rewriting the last 2 sentences.

We removed the clutch concept from the introduction and rewrote the last two sentences.

5. In order to measure tail currents, voltage steps to +40 mV are performed (and no activity is observed). What is the reason for stepping to +40 mV instead of 0 mV, which the authors use as the holding potential?

+40 mV was used to measure the tail currents for GV curves. There would be no tail current at 0 mV in symmetrical solutions. We have clarified this in the Method section.

6. The structural conformations with respect to the C-linker as discussed in Figure 6 are hard to appreciate in the side-by-side representation. The authors could consider adding a third column and show overlays of one subunit in the respective states to visually relay the differences and their interpretation more clearly.

We have added a new supplementary figure (Fig. S8) with pairwise overlays of the four states. We think that these overlays, together with the supplementary movies, visually relay the differences in the structures of the states.

7. On page 9 and 11, the interpretation that the CNBD undergoes a counterclockwise rotation upon cAMP binding is interesting. Can the authors be more specific about how this compares with what was seen in TAX4 upon cGMP binding? Is it the same direction of rotation or opposite? How do the magnitudes of rotation compare? The authors need to provide some rationale as to why spHCN may be different. A figure would also be useful with overlays between spHCN and TAX4 apo to agonist-bound (resting).

We have clarified this in the text and compared the rotation with that in TAX4 in the discussion. We have also now included the .pdb files for each of the four states as supplementary material so they are available for readers to make their own measurements/comparisons.

8. In light of that, the authors should present measurements, as shown in Figure 4, also for the structural models obtained from their Rosetta modeling. Is it possible that in the activated/inactivated states the

signal from different subunits contribute predominantly to tmFRET? In other words, how do the authors make sure, that the observed tmFRET signal reports on the same donor/acceptor pair in all functional states?

As suggested, we have added a new table (Table S1) summarizing the distances between adjacent, same, and diagonal subunits for all tmFRET pairs for all four models. Distances longer than 25 Å would produce negligible FRET. We have, therefore, added a statement in the discussion that the tmFRET is predominantly coming from the adjacent subunits for all states.

9. The differences with HCN1 are really a most intriguing part of this manuscript. Discussion page 12, the authors state that “this clutch might be always disengaged so that HCN1 channels do not inactivate”. Can you provide a framework for this statement? How do you envision the clutch being disengaged in a structural way?

We have further clarified this in the discussion. We think the clutch is disengaged in mammalian channels, so there is no inactivation in mammalian HCN channels. We speculate that the A' helix of mammalian channels might rotate differently in the absence of cAMP, so that it is not positioned to couple the movement of the S4 to channel inactivation.

Minor points:

The title could be improved. It says very little. “Noncanonical” is vague and not particularly appropriate here, since there is no “canonical” mechanism for spHCN inactivation

As suggested, we have changed the title to “Electromechanical coupling mechanism for activation and inactivation of an HCN channel”.

Page 4, 6th line from bottom, should be Fig. S2b instead of Fig.2b

Fixed. Thanks!

Page 6, clarify if the cysteine-reduced spHCN works like WT.

We have added this clarification and referenced our previous paper.

On page 7, second to last paragraph: “There results suggest...” probably should say These results suggest...

Typo fixed. Thanks!

On page 11, last paragraph: “The A' helix have been shown...” probably should be The A' helix has been shown...

Done

In all supplementary figures the respective voltages (holding, hyperpolarization, tail) should be indicated in the panels.

Done

Reviewer #1 (Remarks to the Author):

The authors have addressed all my concerns.

Reviewer #2 (Remarks to the Author):

The authors have addressed all concerns.